# Insights into the Potential of Hardwood Kraft Lignin to Be a Green Platform Material for Emergence of the Biorefinery

**DOI:** 10.3390/polym12081795

**Published:** 2020-08-11

**Authors:** Juliana M. Jardim, Peter W. Hart, Lucian Lucia, Hasan Jameel

**Affiliations:** 1Department of Forest Biomaterials, North Carolina State University, 2820 Faucette Dr. Campus Box 8005, Raleigh, NC 27695, USA; jmjardim@ncsu.edu (J.M.J.); lalucia@ncsu.edu (L.L.); jameel@ncsu.edu (H.J.); 2WestRock, 501 South 5th Street, Richmond, VA 23219, USA; 3Department of Chemistry, North Carolina State University, 2620 Yarbrough Dr. Campus Box 8204, Raleigh, NC 27695, USA; 4State Key Laboratory of Biobased Materials & Green Papermaking, Qilu University of Technology/Shandong Academy of Sciences, Jinan 250353, China

**Keywords:** hardwood, lignin extraction, lignin modification, valorization, chemicals, materials

## Abstract

Lignin is an abundant, renewable, and relatively cheap biobased feedstock that has potential in energy, chemicals, and materials. Kraft lignin, more specifically, has been used for more than 100 years as a self-sustaining energy feedstock for industry after which it has finally reached more widespread commercial appeal. Unfortunately, hardwood kraft lignin (HWKL) has been neglected over these years when compared to softwood kraft lignin (SWKL). Therefore, the present work summarizes and critically reviews the research and development (R&D) dealing specifically with HWKL. It will also cover methods for HWKL extraction from black liquor, as well as its structure, properties, fractionation, and modification. Finally, it will reveal several interesting opportunities for HWKL that include dispersants, adsorbents, antioxidants, aromatic compounds (chemicals), and additives in briquettes, pellets, hydrogels, carbon fibers and polymer blends and composites. HWKL shows great potential for all these applications, however more R&D is needed to make its utilization economically feasible and reach the levels in the commercial lignin market commensurate with SWKL. The motivation for this critical review is to galvanize further studies, especially increased understandings in the field of HWKL, and hence amplify much greater utilization.

## 1. Introduction

Lignin is the main source of renewable aromatic structures on Earth [1,2]. The lignocellulose biorefinery is an interesting approach to maximize the value of lignin and its products. Lignin valorization for industrial applications depends on the development of novel technologies and approaches to overcome challenges mitigating conversion to valuable products [3]. Technical challenges include the need for efficient biomass pretreatment, lignin separation and purification technologies that can address lignin’s diverse structure and complex chemistry. Despite recent advances, there is currently limited conversion of hardwood kraft lignin (HWKL) products into bulk or fine chemicals. Difficulties with the production of value added products from lignin can partly be attributed to the highly complex and condensed nature of kraft lignin, the prevalence of highly recalcitrant lignin linkages/bonding motifs, and a high amount of polydispersity in addition to a considerable sulfur content [4].

Lignin can be utilized as an eco-friendly alternative to numerous petro-derived substances. Potential high-value products from isolated lignin include low-cost carbon fiber, engineering plastics and thermoplastic elastomers, polymeric foams and membranes, and a multitude of fuels and chemicals all currently sourced from petroleum [5]. A multitude of lignin applications have been explored at research level. Although current industrial products are primarily produced from lignosulfonates and sulfonated kraft lignin such as dispersants, emulsifiers, raw material for vanillin and dimethyl sulfoxide (DMSO) production and road-dust suppression agents [6]. Kraft lignin production increased by 150% from 2014 to 2018, competing with lignosulfonates to some extent [7]. The kraft pulping process dominates the industry because of advantages in chemical recovery and pulp strength. Sulfite pulps are not as strong, and the efficiency and effectiveness of the process is more sensitive to wood species characteristics than is the kraft pulping process. Thus, the use of sulfite pulping has declined in comparison to kraft pulping [8].

Kraft mills account for ~two-thirds of the global pulping industry [9] where ~78 million tons of lignin are generated annually [10]. Lignin is present in black liquor (BL) which is typically concentrated by evaporation before combustion in a recovery boiler to produce steam and a majority of the energy necessary for pulping operations [11]. In the biorefinery concept, rather than just producing steam and energy, a portion of the black liquor would be utilized as a feedstock for production of value-added products. Removal of such a feedstock could be used for increasing pulp production since many recovery boilers are limited by total solids loading or steam generation (circulation limit), and hence they represent the capacity-limiting step in several pulp mills. Lignin isolation from black liquor is considered a potential method for reducing these process limitations [12]. It has been estimated that the total amount of lignin produced by the pulp industry is 60% more than what is actually needed for supplying internal power [13].

Although burning lignin is still a valuable contribution for saving fossil sources, processing lignin into value-added applications is a key factor for creating economically feasible biorefinery processes [14]. Lignin is an outstanding feedstock for different materials because of its varied functional groups, aromaticity, renewability, biodegradability, nontoxicity, and relatively low cost. Yet, a gap exists between the kraft lignin producers and market off-takers, due to the current lignin processes having difficulties meeting stringent quality control constraints and market specifications which end-uses require. These constraints slow down lignin market penetration [7].

The literature on the relation between the structure/chemistry of the lignin precursor and the resulting materials’ properties is scarce. Since lignin is not a well-established chemical, but inherently a loosely defined precursor, it is crucial to characterize the lignin used as a precursor to better exploit its structural and chemical features. Every lignin is different and neglecting such a fact gives rise to a plethora of non-comparable works and irreproducible materials portfolios which substantially hinder progress towards lignin-based commercial products [15]. This review is specifically focused on hardwood kraft lignin (HWKL) and critically discusses its structure and properties, as well as extraction, fractionation, and modification processes. Finally, HWKL applications are addressed.

## 2. Lignin Extraction

The separation of lignin from the black liquor generated during the kraft pulping process can be achieved by acidification and ultrafiltration membranes. The acidification technique is based on the dissociation equilibria of weak acid groups which affects the solubility behavior of lignin-related chemical species. Membrane separation processes, on the other hand, can be efficient and cost-effective in many cases, however there are two key limitations: first, such processes get increasingly difficult to control as the concentration of the retained material increases. Second, the flux of permeate passing through a membrane tends to fall during continued usage due to such fouling phenomena as pore plugging and cake formation [16].

LignoBoost and LignoForce processes are the two main commercial technologies for precipitating lignin from black liquor [7], and both can be used to obtain hardwood kraft lignin. The LignoBoost process uses dissolved carbon dioxide (CO_2_) to decrease the pH of the process stream from ~13 to 10 (Figure 1). It is well known that when the BL is acidified, the phenoxide groups of the dissolved lignin become protonated and lignin solubility decreases, i.e., the lignin precipitates. After precipitation, the solids are separated by filtration, then subsequently re-suspended in water and sulfuric acid (H_2_SO_4_) to obtain a lower pH of ~2.5 to remove impurities [17].

At pH values of less than 11, significant quantities of total reduced sulfur (TRS) compounds and other volatile sulfur species can be generated. Such compounds are strongly odorous with well-known negative effects on human health and other forms of life. Thus, LignoForce was created to address these issues [18]. LignoForce is a commercialized technology that first oxidizes the black liquor with O_2_ and then acidifies to pH ~ 9 with CO_2_ [19] (Figure 2).

It is worth mentioning that lignin needs to be precipitated from pulping spent liquors effectively and selectively to have an economically feasible lignin production [19]. Unfortunately, acidic conditions used in lignin precipitation are known to cause some β-ether cleavage and lignin condensation [20]. Thus, the process conditions for lignin recovery from black liquor may interfere with its processing and use. The cleavage and condensation reactions are depicted in Figure 3.

The use of extracted lignins rather than whole biomass can claim an advantage in that the material can be fully dissolved in organic solvents, facilitating recovery and continuous processing [20] for diversification of products and hence value creation for the pulp and paper industry. However, it is important to mention that lignins can be extracted from the whole biomass. Lignins, in any form, are soluble in ionic liquids (ILs), which facilitates extraction from lignocellulose. The extraction can be performed with or without dissolution of the biomass [21]. ILs are considered green solvents due to their non-volatility and low flammability. Furthermore, ILs are not only used as solvents but also play an important part in catalytic cycles in pulping reactions [22]. However, ILs have a major shortcoming as they are much more expensive when compared to common and traditional solvents. Thus, recoverability of ILs should be explored and emphasized. Due to the π–π interaction between ILs and lignin, removal of lignin from ILs is proven to be a complex process and, therefore, requires multiple steps [23]. This makes the recycling and regeneration of ILs, particularly in extremely large volumes, equally cost inefficient [24].

Moreover, deep eutectic solvents (DESs) has been reported to completely isolate lignin from lignocellulosic biomass in a one-pot procedure [25]. DESs are green and inexpensive solvents, that have emerged at the beginning of this century to overcome the problems of ILs [26]. Similar to ILs, DESs have interesting properties including negligible volatility, non-flammability, and high conductivities [27].

## 3. Structure and Properties

Lignin may be described as a complex, racemic, cross-linked, and highly heterogeneous aromatic macromolecule based on hydroxycinnamyl alcohol monomers with various degrees of methoxylation (*p*-coumaryl, coniferyl and sinapyl alcohol) [28]. Lignin building blocks are shown in Figure 4. These structural building blocks are zipped together by ether linkages such as β-O-4, 4-O-5, and by carbon–carbon bonds such as 5–5, β-5, β-1 and β–β linkages [29].

Hardwood lignin is made of coniferyl alcohol and sinapyl alcohol, forming the macromolecules guaiacyl (G) and syringyl (S), respectively. Softwood lignin contains only guaiacyl units in which one of the ortho-positions next to the phenol is free and hence more easily branched and/or crosslinked. In contrast, hardwood lignin that contains both guaiacyl and syringyl units, has a more linear structure compared to softwood [30]. Softwoods lignin appears to vary little amongst species [31,32], whereas hardwood lignins vary greatly between species with a wide range of syringyl-to-guaiacyl ratio (S/G ratio) [33] within its structure.

Native lignin undergoes extensive chemical transformation during the pulping processes, such as fragmentation reactions, ring opening, and sulfonation reactions that facilitate its solubilization in black liquor. The extent of structural modification depends on the severity of the process, i.e., alkali charge, temperature and duration of cooking [34]. Delignification leads to the modification of the frequency of interunit linkages (β-O-4 and C-C linkages), making changes in the frequency of the functional groups, like hydroxyl, carbonyl, and carboxyl groups [14]. The delignification reactions increase the number of phenolic hydroxides due the breakage of the β–aryl bonds and increase the number of condensed C–C bonds [35].

In addition, the hardwood kraft lignin properties depend upon the feedstock from which they are obtained (wood species), the extraction method employed, and downstream purification processes [33]. The depolymerization of hardwood lignin generally results in high aromatic monomer yields because these feedstocks typically contain high S/G)ratios [36].

### 3.1. Linkages

The most frequent lignin inter-unit linkage is the β–O–4- (β-aryl ether linkage), which is the linkage most easily cleaved chemically, providing a basis for industrial processes and several analytical methods [37]. The other linkages are all more resistant to chemical degradation. Table 1 shows the frequency of these linkages in the biomass and Table 2 shows the frequency of linkages for different HWKL.

### 3.2. Molecular Weight

Molecular weight (MW) is an important chemical property for the characterization and utilization of lignin [40]. Reactivity and physico-chemical properties of lignins are partly governed by this property [41]. For these reasons, MW is addressed in most of the papers dealing with lignin. Table 3 depicts the molecular weight distribution of several hardwood kraft lignins.

The non-uniformity of lignin precludes the characterization of a specific molecular weight [50]. Therefore, it is necessary to characterize lignin in terms of average molecular weight. Two common averages used are number average molecular weight (M_n_) and weight average molecular weight (M_w_). The polydispersity index (PDI), defined as M_w_/M_n_, represents the molecular weight distribution of the polymers [51]. The magnitude of M_w_ and M_n_ results depends on the technique used for their determination; overall, values from light scattering are much larger than those obtained from conventional protocols (HPLC and GPC) [40]. In some instances, the expected range of molecular weight will dictate the technique used to measure the molecular weight. When the same sample is subjected to different measurement techniques, it is possible to obtain differing answers. 

### 3.3. Thermo-Mechanical Properties

Lignin is moderately stable at elevated temperatures due to its highly aromatic backbone [52]. Lignin thermally decomposes over a broad temperature range, because various oxygen functional groups from its structure have different thermal stability. Thus, their scission occur at different temperatures. Furthermore. due to its complex composition and structure, the thermal degradation of lignin is strongly influenced by its nature, moisture content, reaction temperature, degradation atmosphere, and mass transfer processes which have a considerable effect on conversion and product yields, as well as on the physical properties and quality of the pyrolysis products [53]. In a conventional thermogravimetric analysis (TGA) curve, main weight loss starts ~250 °C and continues to 600 °C. An inflection point of the main degradation step appears at 382 and 365 °C for softwood kraft lignin (SWKL) and HWKL, respectively [54]. The same researchers evaluated Modulated TGA (MTGA) to study the kinetics of lignin pyrolysis. In this technique, a sinusoidal temperature wave is applied to a sample to calculate and display the kinetic parameters of decomposition on a continuous and real-time basis. Using a constant heating rate, the apparent activation energy (Ea) values were ~230 and ~210 kJ/mol for SWKL and HWKL, respectively. 

Glass transition temperature (T_g_) varies widely depending on the method of lignin isolation, adsorbed water, molecular weight and thermal history [55]. T_g_ of isolated lignin samples is often difficult to determine due to their heterogeneous structures and broad molecular weight. In addition, T_g_ shifts to higher temperatures with increasing average molar mass [56]. Furthermore, it is well recognized that solubility generally follows T_g_, with solubility (in organic solvents) increasing as T_g_ declines [57]. In addition, large amounts of phenolic hydroxyl groups contribute towards higher T_g_ through the formation of intra-molecular hydrogen bonds, thereby creating a physically cross-linked structure [52]. 

Kraft lignins do not undergo a distinctive glass-to-rubber (or fluid) transition when heated [58]. It has been reported that the T_g_ for eucalyptus kraft lignin is around 133 °C. Although, the authors claim that ~ 147 °C, all chains of its amorphous structure will acquire mobility [44]. For other hardwood kraft lignins, T_g_ values such as 108 °C [45] and 138 °C [59] have been reported.

The aromatic structure of lignin provides rigidity and stiffness to materials [60]. Moreover, lignin has been found to be incompatible with most aliphatic polyesters, thus deteriorating the mechanical properties of any resultant composites besides contributing to rougher and fractured surfaces [61]. Mechanical testing of 100% lignin fibers has revealed tensile strengths of 23 MPa and a Young’s modulus of 3.9 GPa [62] These values are comparable to those reported for modified softwood kraft lignin thermoplastic (25 MPa and 1.5 GPa, respectively) [63].

### 3.4. Functional Groups

Functional groups in lignin include methoxyl-, carbonyl-, carboxyl-, and hydroxyl- linking to aromatic or aliphatic moieties [64]. The amount and proportion of these functional groups will depend on the raw material, pulping conditions, and lignin extraction method. Table 4 shows the functional groups reported in the literature for different HWKL.

### 3.5. Reactivity

HWKL reactivity depends on species, delignification method and its severity. These parameters influence the type and amount of functional groups, as well as key lignin linkages, S/G ratio and hence affect lignin reactivity [35,68]. It should be noted that the different functional groups have different reactivities. The amount of hydroxyl groups (-OH), particularly phenolic-hydroxyls, is one of the most important parameters in lignin because it is an indicator of lignin reactivity and give lignin its potential to be utilized in a variety of technical applications [69].

The degree of condensation is an important lignin characteristic as it is often negatively correlated with lignin reactivity [70]. Lignin condensation is a common phenomenon in various strategies of lignin processing/conversion. The increase of condensed structures largely reduces product yields since C-C bonds are more difficult to further decompose into aromatic monomers than structures connected by ether bonds [71]. Figure 5 gives a simplified representation of the condensation process leading to a dead-end product for downstream processing.

The β-O-4 linkage and the degree of condensation in lignin are closely related to the S/G ratio [33]. The C-5 position of the syringyl aromatic ring is occupied by a methoxyl group and therefore it cannot be involved in condensation. Thus, hardwood lignins are less condensed than softwood lignins [39], hence more reactive. A high lignin isolation yield usually corresponds to low reactivity towards depolymerization, and vice versa [35]. Therefore, hardwood lignins are expected to show lower isolation yields than softwoods, which has been reported elsewhere [72].

### 3.6. Elemental Composition

Table 5 shows the elemental composition of different hardwood kraft lignins. The differences in composition are related to their origins and to extraction processes [73]. The high carbon content makes HWKL very attractive as a source to a wide range of materials. Moreover, kraft lignin contains sulfur (S) and ash, which might limit its applications. Overall, an ideal kraft lignin has very low amounts of these components.

## 4. Lignin Fractionation

One of the main issues limiting the application of technical lignins is their heterogeneity and high polydispersity [1]. Various types of C–O and C–C linkages with broad bond dissociation energy (BDE) linked between the aromatic centers of lignin lead to low selectivity in lignin depolymerization. Moreover, functional groups have an impact on lignin reactivity as previously discussed in § “Reactivity”, and hence they affect lignin conversion. Lignin’s relatively high molecular weight and amorphous structure results in limited solubility in most common organic solvents at standard temperature and pressure (STP). In addition, the high reactivity of lignin degraded intermediates which are prone to side reactions makes it very difficult to produce a high yield of desired products from lignin [3].

Fractionation based on size and/or composition is recommended for certain applications. A good fractionation method isolates lignin with high yield, high purity, and retains β-O-4 bonds in the technical lignins, which would be beneficial for subsequent conversion processes [3]. Fractionation enables development of a multi-stream integrated biorefinery since it creates multiple lignin streams with uniformity and specific chemical characteristics. These fractionated lignin streams could be used as starting materials for multiple biomanufacturing streams to make different value-added products [75]. Three common fractionation methods are fractionation with organic solvents, ultrafiltration, and acid precipitation.

### 4.1. Fractionation with Organic Solvents

Typically, this method starts from a state of complete dissolution of lignin in a polar solvent, from which specific lignin fractions can be precipitated upon gradual addition of a selected miscible but nonpolar solvent. In this way, the molecular composition of each lignin fraction can be manipulated by adjusting the solvent mixture along an almost continuous solvent gradient [76]. This fractionation method has been widely used on an academic research level; however, the scale up of this technique seems compromised by the use of large quantities of organic solvents [1]. Fractionation of hardwood and softwood kraft lignins using dichloromethane, propanol and methanol has been reported. The solubility of HWKL was much higher (81%) than SWKL (34%) [77]. 

### 4.2. Fractionation with Ultrafiltration Membranes

Ultrafiltration is another fractionation technique, which uses a series of membranes that allows for a very efficient control of the molecular mass distribution of lignin fractions by the choice of different membrane cut-offs [78]. Ultrafiltrated lignin fractions are less contaminated with carbohydrates compared to precipitated lignin [79]. This method has been shown to be effective for the fractionation of eucalyptus black liquor. The lignin fractions were heterogeneous not only with respect to their molecular weight, but also with respect to the type and frequency of functional groups [80]. The decreasing molecular size was followed by a decrease in total phenolic and aliphatic hydroxyl groups. Also, the increase in membrane cut-off molecular weights led to fractions with a higher solids content and lower amounts of inorganics. Moreover, the highest content of non-condensed structures, i.e., highest content of β-O-4 structures (23 per 100 aromatic rings), was observed in the fraction with the smallest molecular weight.

### 4.3. Fractionation by Acid Precipitation

Acid precipitation is a common and simple method to obtain lignin fractions that uses strong acid to gradually decrease the pH of black liquor [81,82,83,84]. This technique is based on the dissociation equilibria of weak acid groups which affect the solubility behavior of lignin-related chemical species. Different pH intervals generate lignin fractions with different yields, composition and properties [72,81,82].

Acid precipitation has been shown to be an efficient method for fractionating hardwood and softwood kraft lignins into molecules with different relative molecular masses [83]. Typically, the lignin fractions precipitated at higher pH show higher molecular weights, and as pH is lowered, lignin fractions with smaller molecular weights are precipitated.

## 5. Lignin Modification

Although lignin presents potential direct applications in the polymer industry, it can only be incorporated in small amounts, considering its thermal degradation and mechanical properties. The modification of lignin seems to be a way to overcome the limitations, using this renewable product as a starting material for polymer and chemical synthesis [1,56].

Different types of modification have been proposed to increase its chemical reactivity, reduce the brittleness of lignin-derived polymers, increase its solubility in organic solvents, and improve the ease of processing the lignin. These modifications consist of increasing the reactivity of hydroxyl groups or changing the nature of chemical active sites, but it is always with a view to synthesizing new, efficient, and more reactive macromonomers [56].

Modifications such as oxidative pre-treatment of hardwood kraft lignin has been reported to introduce oxygen in the material and to induce specific structural changes leading to much more stability against thermal degradation [85]. The oxidative pretreatment is performed in air, at temperatures around 250 °C, and at slow heating rates, which leads to an increase in oxygen content mostly due to carbonyl incorporation. There are several modification techniques reported in the literature, however the present review only addresses the most common and relevant methods to hardwood kraft lignin.

### 5.1. Sulfomethylation or Sulfonation

The supply of lignosulfonate, lignin obtained from the sulfite pulping process, is limited by the small number of sulfite pulping processes currently operating in the world, but there is demand for it due to its water-solubility. Kraft lignins are not water-soluble; however, this can be achievedvia sulfomethylation or sulfonation [86].

Sulfomethylated hardwood kraft lignin has been successfully prepared in an aqueous medium by using sodium sulfite and formaldehyde under alkali conditions as shown in Figure 6; the sulfomethylated lignin is formed via electrophilic substitution of the ring by a sodium methylsulfonate group [86].

During sulfomethylation, introduction of sulfonate and methyl groups into lignin takes place, which leads to an increased water-solubility and an increase in thermal stability [86]. In addition, the lignin modification increases molecular weight of the lignin due to the replacement of hydroxyl groups with sulfomethyl groups and possibly due to condensation reactions under alkaline conditions. Lignin extracted from birch has been converted to a water-soluble product via sulfonation with sodium sulfite under alkaline conditions [87].

### 5.2. Alkylation

Due to the presence of a very large amount of phenolic hydroxyl groups in kraft lignins, alkylation can be effortlessly carried out stoichiometrically through nucleophilic aromatic substitution reactions. These are well-acknowledged derivatization reactions that produce lignin ethers and hydroxyalkyl lignins which are used to prepare polymer blends for mixing with synthetic polymers [52].

Alkylation of hardwood kraft lignin by bromododecane (C_12_H_23_Br) is shown in Figure 7. This reaction is reported to improve the compatibility of HWKL when blended with polypropylene [88].

Alkylation completely removes the possibility of radical initiated self-polymerization of the kraft lignin, making it an important step for commercial processing of this natural polymer [89]. However, after alkylation, the phenolic hydroxyl groups are converted to less acidic secondary aliphatic hydroxyl groups, which eventually reduce the extent of intra-molecular hydrogen bonding and thereby reduces T_g_ [52].

### 5.3. Methylation

Methylation is generally achieved by reacting lignin with diazomethane or dimethyl sulfate, as shown in Figure 8. Both reagents are highly toxic, which constitutes a major drawback when considering the use of lignin in a more global green chemistry effort [1].

After methylation of phenolic hydroxyl groups, a lower degree of hydrogen bonding between individual lignin macromolecules should be expected, which should facilitate the macro-Brownian motion of lignin and thus its fusibility [85]. Also, the modified lignin form miscible blends with various aliphatic polyester.

Another research effort verified that methylation of HWKL using dimethyl sulfate is selective toward the phenolic hydroxyl groups, allowing for a progressive methylation that can be adjusted by varying the feed ratio of dimethyl sulfate to the phenolic hydroxyl groups. The hardwood kraft lignin seemed to be relatively less reactive than softwood, which was attributed to the relatively low reactivity of phenolic hydroxyl groups present in HWKL [90].

### 5.4. Esterification

Lignin esterification has been studied as a strategy to improve compatibility between lignin and polyolefins in blends and composites [46,65,91,92]. Figure 9 shows the esterification of lignin by acetic anhydride. 

Esterified lignins from different sources, including HWKL (*Eucalyptus* spp.), have been investigated in lignin–polyethylene blends. The modification rendered purer lignin derivatives and less polar, as sugar and ash content were decreased, and hydroxyl groups were completely esterified [48]. The researchers also mentioned that modified lignins exhibited better miscibility with polyethylene than unmodified lignins.

### 5.5. Carboxymethylation

Carboxymethylation of mixed hardwood kraft lignin has been performed using sodium chloroacetate as a carboxylate group donor (Figure 10). Under alkali conditions, NaOH reacts with the hydroxyl group of the lignin’s aromatic ring, generating a strong nucleophile. The alkoxide ion from the alkali lignin attacks the chloroacetate via an SN_2_ reaction resulting in the ether methyl carboxylation of lignin. The degree of carboxymethylation depends on the number of hydroxyl group substitutions with carboxymethyl groups [93]. Carboxymethylated lignins are water soluble and show a relatively high charge density and low molecular weight, features that allow for the utilization as dispersant, for instance. 

### 5.6. Grafting

Grafting attaches polymer chains to the lignin hydroxyl groups, thus resulting in a star-like branched copolymer that possesses a lignin core. Two distinct routes can be used for the preparation of graft copolymers, and both have been applied to the manufacture of lignin graft copolymers, as depicted in Figure 11 [1]. “Grafting from” (Figure 11a): lignin plays the role a macro-initiator for the polymerization. A monomer first reacts with the lignin hydroxyls, and then the polymerization starts. The polymer chain is built on the lignin core. In the case of “grafting onto” (Figure 11b): the polymer chain is first synthesized, and functionalized at one end, to be able to react with the lignin OH functions. The polymer chains are then grafted to the lignin core [1].

Covalent grafting of a maleyl chain on HWKL and on other lignin types in acidic ionic liquid have been reported to be feasible [65]. Lignin esterification of industrial lignins with maleic anhydride (C_2_H_2_(CO)_2_O) was shown. Although the esterification induced a slight decrease in thermal stability, the modified lignin remained compatible with temperature conditions of extrusion processes for the generation of partially bio-sourced composites. Furthermore, the reaction significantly increased lignin solubility in polar and protic solvent, probably due to the additional availability of carboxylic groups resulting from mono-acylation.

## 6. Lignin Applications

Major research and development findings for the applications of hardwood kraft lignins are addressed within this section.

### 6.1. Briquettes and Pellets

The interest in renewable energy sources (RES) has been increasing worldwide as evidenced by the massive interest in pellets and briquettes. These two materials are fuels manufactured from biomass. European countries consumed 50% of the global wood pellets in 2018 [94]. Furthermore, the United States manufactured 8.2 million tons in the same year; being the second largest producer in the world only surpassed by China. It should be noted that the market continues to grow due to high demand from overseas markets.

Direct addition of 6% (w/w) of HWKL to briquettes has been reported to positively influence apparent density, energy density, heating value, and mechanical resistance of the material [95]. Moreover, the addition of HWKL to pellets has been reported to be feasible. Improvements in physical and mechanical properties (density, mechanical durability, fines, and hardness) have been observed [96]. This study highlighted the importance of a lignin with low ash and moisture contents for briquette and pellet applications.

Although this application is still at the research level, the previously mentioned promising results point to the high potential of briquettes and pellets to supply part of the energy consumed around the world, with a vast potential utilization curve for hardwood kraft lignin without concomitant fractionation and/or modification.

### 6.2. Dispersant

Dispersant is a term generically used to describe surfactants, plasticizers, or emulsifiers, depending on the field of application. Healthcare, food, civil construction, and agriculture greatly benefit from dispersants that enable the mixing of immiscible liquid phases and enhance the stability of particle suspensions. Dispersants lower the interfacial tension between immiscible liquids, as well as increasing the repulsive forces between suspended particles and prevent settling and aggregation of phases, thus improving technical properties of multiphase systems such as rheology, lifetime, and function [97].

As discussed in previous sections, kraft lignins are not water-soluble. Thus, to use HWKL as a dispersant, modification is required. A study of carboxymethylated hardwood kraft lignin shows that it can successfully be used as a dispersant for clay suspensions [93]. The researchers added that it could potentially be applied in pesticide formulations, ceramic suspensions, and as a cement admixture. The optimal conditions for carboxymethylation were 1.5 M NaOH concentration, a 3 mol/mol sodium chloroacetate (SCA)/lignin ratio, 40 °C, 4 h and 16.7 g/L lignin concentration. In addition, this modified lignin had a charge density and carboxylate group of 1.8 meq/g and 1.68 mmol/g, respectively.

In another study, sulfometylated HWKL was produced with formaldehyde and sodium sulfite under alkali conditions. The optimum conditions for the lignin modification were 0.5 M NaOH(aq), 0.9 mol/mol sodium hydroxymethyl sulfonate/lignin at 100 °C for 3 h, and 20 g/L lignin concentration [86]. It was shown that the modified lignin had a charge density of −1.60 meq/g and sulfonate group content of 1.48 mmol/g. The sulfomethylated lignin was used as a cement dispersant, and the dispersibility of cement was increased from 60 to 155 mm by adding 1.2 wt% of modified lignin to cement. The researchers also evaluated the addition of unmodified lignin, which did not change the dispersibility of cement.

The majority of the industrially used dispersants are synthesized from non-renewable precursors and are not biodegradable, raising concerns over their sustainability [97]. Therefore, the development of lignin-based dispersants is an attractive solution. Moreover, laboratory experiments have already shown that its manufacturing from modified HWKL is feasible.

### 6.3. Adsorbents

Lignin presents a good capacity to adsorb heavy metal ions because it possesses two types of acidic sites (carboxyl and phenol groups) that participate in the sorption mechanism. Thus, ion exchange using lignin has been studied as a potential low cost method for wastewater purification [1,74]. Eucalyptus kraft lignin has been studied for the removal of Cu(II) and Cd(II) from water/wastewaters in single and multi-component systems [74]. The researchers highlighted a superior performance of HWKL compared to most adsorbents, carbons, and biosorbents currently utilized. It was also mentioned that hardwood kraft lignin as an adsorbent is not commercial yet, however laboratory results show that it can be applied for development of large-scale systems.

Besides ion-exchange, lignin derivatives can efficiently capture metal ions through chelation and electrostatic interactions. A review paper about lignin application as an adsorbent of heavy metal states that lignin can be modified by physical/chemical methods to fabricate desirable adsorbents with good sorption capacity and selectivity for the target metals [98]. The researchers also mentioned that lignin-based materials have shown outstanding sorption for metals such as toxic metals (Hg), precious metals (Ag), and metal anions (Cr). Furthermore, it was recommended that specific emphasis should be placed on the lignin modification to design and develop advanced lignin-based adsorbents.

### 6.4. Hydrogels

Hydrogels are three-dimensional polymeric networks formed from cross-linked hydrophilic polymers. They are insoluble and capable of retaining a large amount of water in their swollen state. They are typically used for contact lenses, hygiene products, wound dressings, drug delivery and tissue engineering. 

Hydrogels synthesis through radical polymerization of hardwood kraft lignin, N-isopropylacrylamide, and N,N′-methylenebisacrylamide is shown in Figure 12.

The reactions involved in the production of lignin-based hydrogels are well described elsewhere [99]. The results of the study showed that lignin-based hydrogels exhibited less swelling affinity because they possessed a reduced surface area and a less porous structure than synthetic hydrogels. On the other hand, they were reported to be more thermally stable. The incorporation of lignin generated a less cross-linked hydrogel which tended to increase the rigidity and rheological stability of the hydrogel. It was also stated that, when compared to synthetic hydrogels, lignin-based hydrogels exhibited less elastic behavior as temperature increased. This is the only study dealing with hardwood kraft lignin in lignin-based hydrogels.

### 6.5. Carbon Fibers (CF)

CF are high-strength light-weight materials and their application in composites takes advantage of their strength, stiffness, low weight, fatigue characteristics, lack of corrosion and heat insulation [11]. The main applications of CFs are in construction, electronics, transportation, and aviation. Currently, carbon fibers are manufactured with polyacrylonitrile (PAN) and pitch, two non-renewable materials. 

One of the key driving forces for the promotion of the CFs market is the potential for lightweight automobiles. However, the high cost (~ $35/kg) of CFs can inhibit their utilization for commercial applications [100]. Lignin-based carbon fibers with their low-cost and sustainability appeal characterize a good alternative for the segment [5]. Furthermore, lignin is expected to offer additional benefits to CF, such as elimination of toxic substances involved in the preparation [101], lower melting temperature and faster stabilization [102] when compared with PAN- and pitch-based CFs.

To obtain lignin-derived CF, isolated lignin is first processed into fibers by extruding filaments from a melt or solvent swollen gel (spinning), and then the spun fibers are thermally stabilized in air where the lignin fiber is oxidized (stabilization). Afterwards, the fibers are subjected to pyrolysis under nitrogen or inert atmosphere, where fibers become carbonized through the elimination of hydrocarbon volatiles, their oxidized derivatives, carbon monoxide, carbon dioxide, and moisture [5]. Figure 13 provides a model workflow for the preparation of carbon fibers from lignin. 

CF from hardwood kraft lignin with mechanical properties suitable for general performance grades have been reported [103]. It was shown that thermally pretreating the lignin to remove volatile contaminants disrupts fiber integrity during subsequent thermal spinning and decreases hydroxyl content and subsequent intermolecular interactions by condensing the lignin aromatic nuclei.

The spinnability of lignin appears to be highly dependent on its structure. Hardwood lignins, whose structure is rather linear, can be melt spun without any additive [1,103]. Blending HWKL with poly(ethyleneoxide) (PEO) produced miscible polymer blends which facilitated thermal spinning [103]. Furthermore, SWKL is reported to have spinning difficulties that can be overcome by addition of HWKL permeate as a softening agent [104].

It has been reported that HWKL can also be successfully transformed into CF by blending it with synthetic polymers such as poly(ethylene terephthalate) (PET) and poly(ethylene oxide) (PP), especially with the former [62]. Both systems were easily spun into fibers and blend composition affected surface morphology of the carbon fibers. Immiscible lignin–PP fibers resulted in a hollow and/or porous carbon fiber, whereas carbon fiber produced from miscible lignin–PET fibers tended to display a smooth surface. 

Production of CF from a HWKL copolymer with PAN has also been reported. The resulting copolymer was confirmed by a Fourier transform infrared (FTIR), ^13^C, and ^1^H nuclear magnetic resonance (NMR) spectroscopy, showing the presence of the C≡N group fromPAN co-eluting with ether, hydroxyl, and aromatic groups that are attributed to lignin. The average tensile strength of the CF was 2.41 gf/den, a tensile strain of 11.04%, and a modulus of 22.92 gf/den [105]. 

Stabilized hardwood and softwood kraft lignin CF have shown a skin-core structure similar to fibers made from pitch [106]. Moreover, pore creation in immiscible polymer blends of hardwood kraft lignin and PP occurs by a two-step process: oxidative degradation of the PP component followed by pyrolysis gasification of residual PP related components. Gasification is the main factor for pore growth. The internal surface area of the lignin-based CF (499 m^2^/g) was lower than that for commercial activated carbons (745 m^2^/g) [107]. However, the researchers assure that relatively simple processes, such as steam activation, could effectively activate these porous lignin carbon fibers and make them suitable for commercial applications.

Finally, a recent article demonstrated the development of activated carbon fiber electrodes produced from HWKL to fabricate electric double layer capacitors (EDLCs) with high energy and power densities using an IL electrolyte [108]. A mixture solution of HWKL, polyethylene glycol as a sacrificial polymer, and hexamethylenetetramine as a crosslinker in dimethylformamide/acetic acid (6/4) was electrospun, and the obtained fibers were easily thermostabilized, followed by carbonization and steam activation to yield activated CF.

### 6.6. Antioxidants

Antioxidant properties of HWKL have been determined by a DPPH (2,2-diphenyl-1-picryl-hydrazyl-hydrate) free radical assay. This technique is based on electron-transfer that produces a violet solution in ethanol. The radical scavenger activity is expressed in terms of the number of antioxidants necessary to decrease the initial DPPH absorbance by 50% (IC50). The inhibitory effect of HWKL samples was ~8.4 µg/mL, whereas the IC50 for commercial antioxidant butylated hydroxytoluene (BHT), ascorbic acid, and Trolox was 13.3 µg/mL, 2.9 µg/mL, and 3.4 µg/mL, respectively [109], which shows the great antioxidant capacity of HWKL.

Overall, antioxidant activity increases with the phenolic hydroxyl content because they can scavenge free radicals, which are reduced with aliphatic hydroxyl content. Lignin with lower molecular mass and narrower molecular weight distribution seems to be beneficial [110], which also show the great potential of hardwood kraft lignins.

Another effort evaluated the antioxidant activity of HWKL by the 2,2′-azino-bis(3-ethylbenzothiazoline-6-sulphonic acid (ABTS) assay, which measures the relative ability of antioxidants to scavenge the ABTS generated in aqueous phase. The HWKL could oxidize ABTS to ABTS+ due to its reducing potential, which resulted in a color change (blue to green) [81].

Kraft lignins have shown potential to act as antioxidant for food, cosmetic and pharmaceutical industries instead of BHT, a synthetic resource. Lignin as a cosmetic or pharmaceutical product, is still not regulated because studies dealing with the safety of its use in humans are needed [109]. Recently, hardwood lignin nanoparticles were reported to enhance sunscreen performance. The best formulation had a UV transmittance of only 0.5–3.8% over the entire UVA−UVB region compared to 2.7–51.1% of a commercial SPF 15 sunscreen [111].

### 6.7. Aromatic Compounds–Chemicals

Lignin valorization in solvent systems to produce renewable aromatic chemicals has attracted great attention during recent years. The methodologies to obtain these compounds can be categorized as hydrolysis, hydrogenolysis, oxidation, and a two-step lignin depolymerization. Moreover, catalysis is a promising technique for lignin depolymerization to specific products [112].

Benzene, toluene, and xylene (BTX) and phenols are high-value chemicals that can be more sustainably sourced from lignin than from fossil–based resources [113]. BTX is the precursor for a series of materials such as resins, nylon fibers, polyurethane and polyester; thus, production of BTX from lignin could expand the utilization of lignin–based materials. It should be noted that when targeting chemicals from lignin, a key objective during the fractionation and depolymerization stage is to minimize lignin condensation as stated earlier in this review [35].

Other aromatic compounds that can be produced from lignin are syringaldehyde and vanillin. Eucalyptus and Northern European HWKL have been investigated to produce these compounds by oxidation with O_2_ in alkaline medium. The total yield of syringaldehyde was 14%, whereas for vanillin it was 16% [114]. In another study of oxygen oxidation of eucalyptus kraft lignin, under optimum conditions, only a reduced number of phenolic aldehydes (4%) was obtained. In contrast, when in the presence of catalysts, the yield could be increased to 14% with nitrobenzene and to 8% with CuO [115]. This was attributed to the low yield of transformation of the lignin/lignin oxidation products into low molecular weight acids. 

Vanillin, the highest volume aroma chemical produced worldwide, is produced from a variety of sources, namely oil (85%), woody biomass (15%), and orchid pods (<1%). Around 20,000 tons of vanillin are produced per year, 15% of which comes from lignin (around 3000 tons/y) [116]. Lignosulfonates are the main sources for its production; however, kraft lignin could also be used for this purpose. The use of lignin by chemistry and polymer industries represents an important field of research with major issues in terms of scientific, economic, and environmental points of view and it seems justified that lignin will become a promising renewable aromatic resource in subsequent years [56]. Syringaldehyde is another promising aromatic aldehyde that possesses worthy bioactive properties which can be used in pharmaceuticals, food, cosmetics, textiles, pulp and paper industries, and biological control applications.

### 6.8. Polymer Blends and Composites

Lignin has been added to several polymers for the express purpose of potentially providing valuable new composite properties. Because of their large number of polar functional groups, lignin molecules interact strongly with each other. Most polymers are immiscible with lignin because of weaker interactions between lignin and the matrix polymer than among lignin molecules. Therefore, competitive interactions determine the structure and properties of blends and composites [68].

The miscibility of synthetic polymers such as poly(ethyleneoxide) (PEO), poly(ethylene terephthalate) (PET) and poly(vinyl alcohol) (PVA) with HWKL has been investigated. Miscible blends were observed in lignin/PEO and lignin/PET blends, while immiscible blends were found in lignin/PP and lignin/PVA blends [45]. The former polymers possess functional groups capable of interacting with lignin through secondary forces-bonding whereas the latter do not (Figure 14).

A possible approach towards lignin valorization is using it as a component in plastics. The development of lignin-based thermoplastics relies on altering the viscoelastic properties of lignin through chemical modification or polymer blending [45]. The latter is a convenient method to develop products with desirable properties. The chemical and physical properties of the blends/composites are dependent on monomer type(s), molecular weight, and distribution and composition of the respective polymers [2,45].

Carbon-neutral thermoplastics have been successfully prepared by copolymerization of modified HWKL (oak variants) and dicarboxy-terminated polybutadiene (PBD-(COOH)_2_). Modified lignin (either by fractionation with methanol or by formaldehyde crosslinking) showed high molecular weight which facilitated preparation of free-standing films of lignin-based thermoplastic [117]. Additionally, it enabled formation of a more continuous network with telechelic polybutadiene, whereas the very broad molecular weight distribution of unmodified lignin formed a poorly networked structure.

Lignin-based thermoplastic miscible blends have been studied using HWKL and PEO. Incorporation of small amounts of PEO (5–10% w/w) sufficiently disrupts the lignin supramacromolecular complexes leading to enhanced physical properties. Increasing PEO incorporation further disrupts the lignin structure and the observed physical properties become more influenced by the PEO component [2].

It has been reported that olefinic thermoplastic polymer compositions comprising at least one polyolefin and eucalyptus kraft lignin, can successfully be manufactured [118]. The patent assures increase in the following properties: flow index (MFI), thermo-oxidative resistance (OIT-oxidative induction time), heat deflection temperature (HDT), stiffness (elasticity module), breaking strength, and flexural strength. In addition, the material maintains hardness and tensile strength measured at the outflow.

Blends of eucalyptus kraft lignin and PBAT (a biodegradable polyester, produced by BASF, and based on the monomers 1,4-butanediol, adipic acid, and terephthalic acid) have been studied. The study showed that the addition of up to 20% of lignin generated miscible or partially miscible structured blends, where lignin acted as a lubricant. Furthermore, the bends reached the technical requirements needed for sustainable solutions to rigid plastic apparatus in the agricultural segment, such as seedling tubes [44].

The intermolecular interactions of HWKL and PVA blend fibers prepared by thermal extrusion have been studied [119]. Although the blend is immiscible (presents two distinct T_g_’s) some of the lignin was closely associated with the PVA in the PVA-rich phase. FT-IR analysis confirmed the formation of a relatively strong hydrogen bond between the hydroxyl groups of the short-chain PVA and lignin.

Petroleum-based polyol was replaced with hardwood kraft lignin preparing rigid polyurethane foam. The prepared foams contained from 9% to 28% (w/w) HKL [120]. The addition of lignin reduced the density of the foams, which is desirable if the foam is used as packing or insulation material. Furthermore, the authors reported that the majority of the kraft lignin was chemically crosslinked and the foams had satisfactory structure and strength up to 23% (w/w) addition when a chain extender was applied.

A polyurethane–lignin copolymer has been produced by step-growth polymerization of modified eucalyptus kraft lignin with isocyanate and doped by multi-wall carbon nanotubes (MWCNTs) [121,122,123]. Lignin possesses ion-exchange properties due to the presence of a variety of functional groups, which makes it an attractive active substance for chemical sensing. Co-polymerization allows the fixing of lignin inside a polymer matrix, ensuring high stability of the resulting material [121]. Furthermore, the electrical conductivity and impedance spectroscopy measurements revealed that the interaction between carbon nanotubes and lignin molecules in the polymer enhances its electrical conductivity [122].

Lignin fractions, having different molecular weights and varied chemical structures from HWKL (*Eucalyptus grandis*), were incorporated into a tunicate cellulose nanofibers (CNF)–starch mixture to prepare 100% bio-based composite films. In general, the addition of lignin led to a decrease in thermal stability and tensile strain and an increase in the Young’s modulus of the composite film [49]. Furthermore, some aggregates were observed in the composite films which may explain the lower tensile stress. The researchers pointed out that hardwood lignin with lower molecular weight and polydispersity reinforced the film structure.

The utilization of lignin in polyolefin blends is of growing research interest. However, the low compatibility of lignin and polyolefins restrains a satisfying blend production and leads to poor mechanical properties. It has been reported that it is possible to overcome these issues by modifying lignin prior to its incorporation into polymers to reduce its polarity [48]. The researchers performed esterification of eucalyptus kraft lignin and subsequently blended it with PE at a weight ratio of 1:1. They found a reduction of volume swelling and weight gain when modified lignin was used, which can be related to the esterification of hydroxyl groups as well as a significant reduction of sugar and ash contamination. Moreover, they observed that the elongation at maximum strength decreased to 22% (εM = 11%), while for other technical lignins the decrease was of ~10%. Highest tensile strengths were observed for the hardwood blends (17 N/mm^2^).

Copolymerization of N-isopropylacrylamide (NIPAM) with HWKL has been achieved by atomic transfer radical polymerization using a selectively modified lignin-based macroinitiator as shown in Figure 15. The thermal decomposition temperature of the lignin-polyNIPAM copolymers are reported to significantly increase with increasing degrees of polymerization of NIPAM. The solubility of the lignin–polyNIPAM copolymers in water was dependent on copolymer structure. In both the water-soluble and suspended copolymers, at temperatures above 32°C, the component has been reported to undergo hydrophilic-to-hydrophobic transition, resulting in precipitation of the copolymer [124].

Finally, lignin and acrylonitrile butadiene (ABS) blends and composites have received great attention recently. ABS is a very popular polymer material as shown by its widespread use as a thermoplastic resin in the automotive, marine, home appliance, toys, and other industries. Lignin is generally incompatible with ABS polymers, as shown by large phase separation domains with poor interfacial adhesion within the ABS matrix. This unfortunately leads to significant reductions in the strength and ductility of the resulting composite/blend, thereby limiting practical utility. A recent patent has addressed those issues and claims that by adding a compatibilizing agent, any lignin type, including HWKL, could be added to ABS plastics to provide increased stiffness and reduce cost [125]. The inventors used compatibilizing agents such as polyalkylene oxide, polyvinyl alcohol, polyvinyl acetate, ethylene vinylacetate copolymer, styrene-maleic copolymers, nitrile rubber, and others to assist in the dispersion and/or distribution and/or miscibility of lignin and ABS.

## 7. Conclusions

Due to lignin’s high-energy density and intrinsic aromatic based structure, this biomaterial is an ideal renewable feedstock that has tremendous potential in defining modern biorefinery. Lignin shows great potential to produce fuels, value-added chemicals, and functional materials, and ultimately reduce the environmental impact of their production. Although several studies have focused on converting lignin to valuable products, only a few of these efforts are commercially profitable, mainly because of low yields and low quality of the final products [3]. It is unfortunate that most lignin work deals with softwood kraft lignin, while hardwoods have been neglected. Because HWKL has a very different structure to SWKL, the knowledge acquired from the latter species cannot always be applied to the former. Thus, the intent of the present review was to provide a good basis for its possible valorization by overviewing the HWKL structure and properties. By appropriate understanding of the concepts in this review, it is hoped that utilization of this vast available and cheap aromatic source will be expanded to promote a firmer footing for a growing biorefinery.

## Figures and Tables

**Figure 1 polymers-12-01795-f001:**
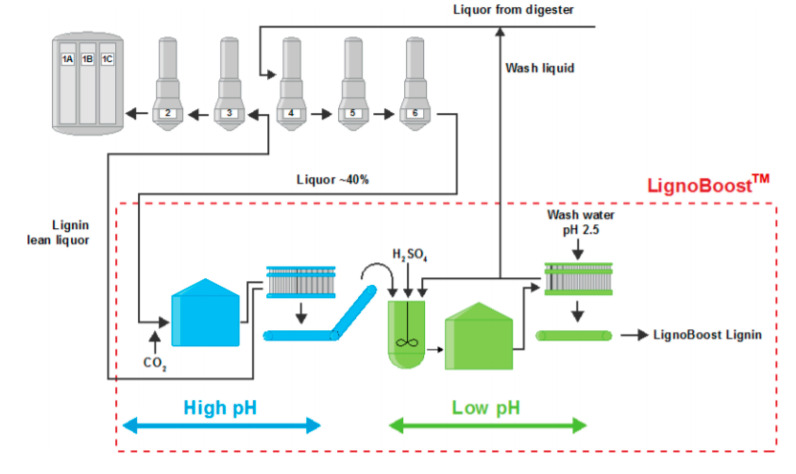
LignoBoost process by Stora Enso.

**Figure 2 polymers-12-01795-f002:**
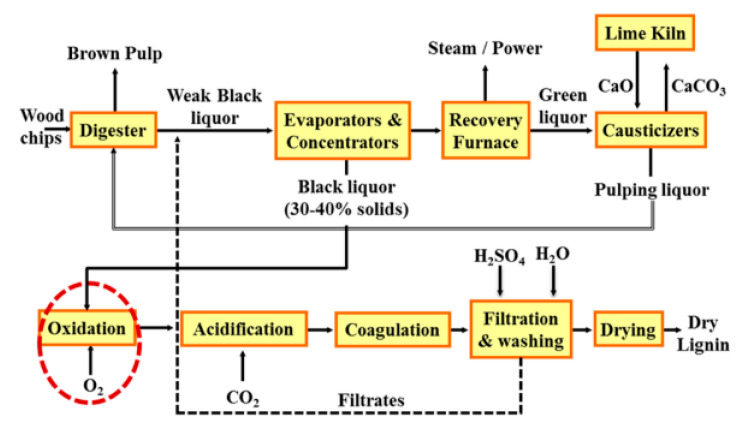
LignoForce System by FPInnovations and NORAM [18].

**Figure 3 polymers-12-01795-f003:**
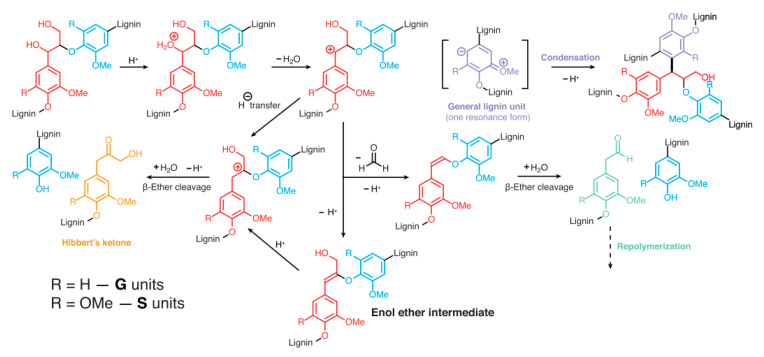
Lignin acidolysis and condensation routes [20].

**Figure 4 polymers-12-01795-f004:**
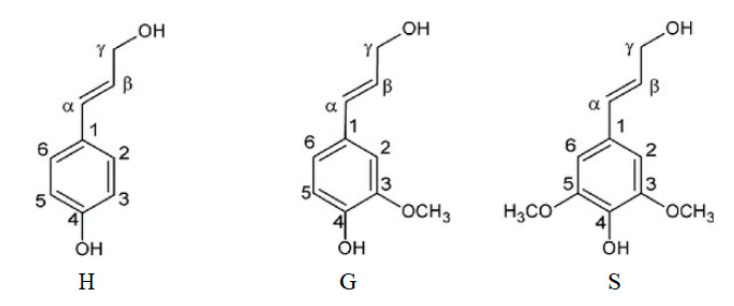
Lignin building blocks: *p*-coumaryl alcohol (H), coniferyl alcohol (G), and sinapyl alcohol (S).

**Figure 5 polymers-12-01795-f005:**
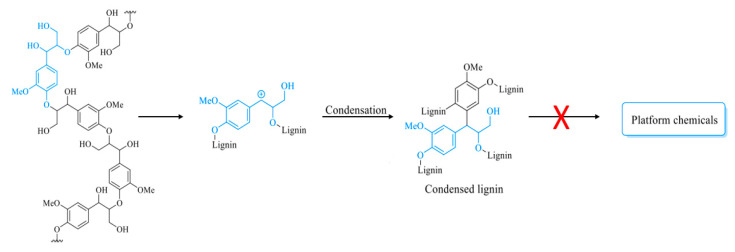
Schematic diagram of lignin condensation.

**Figure 6 polymers-12-01795-f006:**
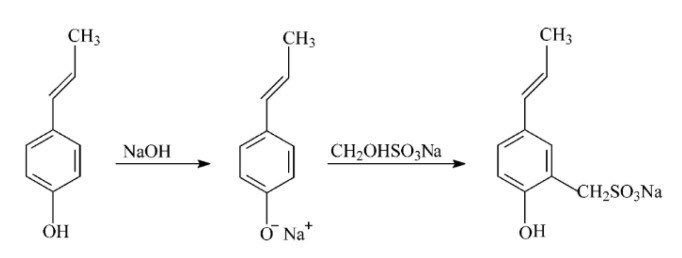
Sulfomethylation of lignin sodium sulfite and formaldehyde [86].

**Figure 7 polymers-12-01795-f007:**
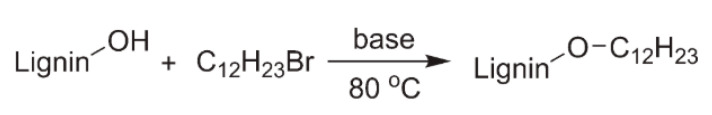
Reaction progress for the alkylation of lignin hydroxyl groups by bromododecane [88].

**Figure 8 polymers-12-01795-f008:**
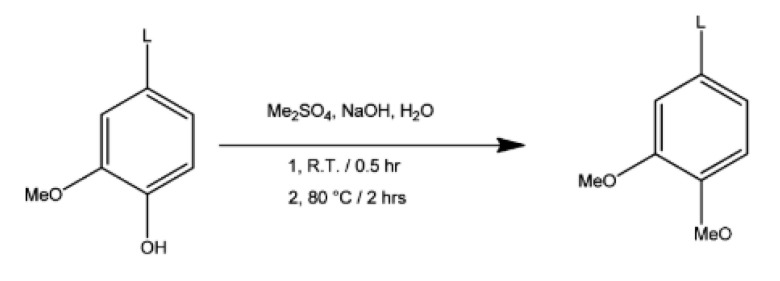
Methylation of kraft lignin by dimethyl sulfate in aqueous NaOH [90].

**Figure 9 polymers-12-01795-f009:**
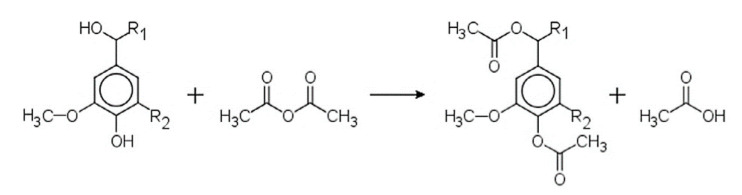
Esterification of lignin by acetic anhydride [68].

**Figure 10 polymers-12-01795-f010:**
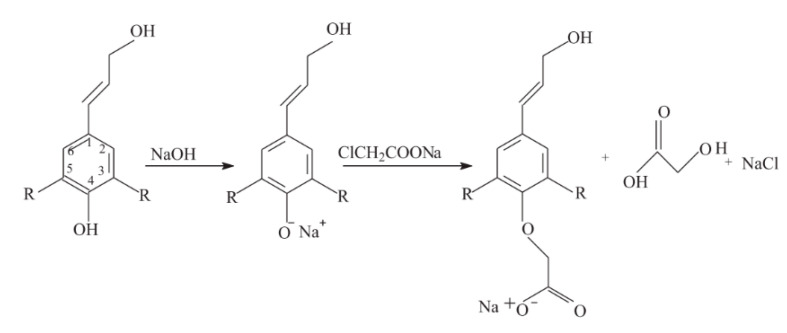
Carboxymethylation of lignin reaction scheme. ‘‘R’’ can be OCH_3_ or H [93].

**Figure 11 polymers-12-01795-f011:**
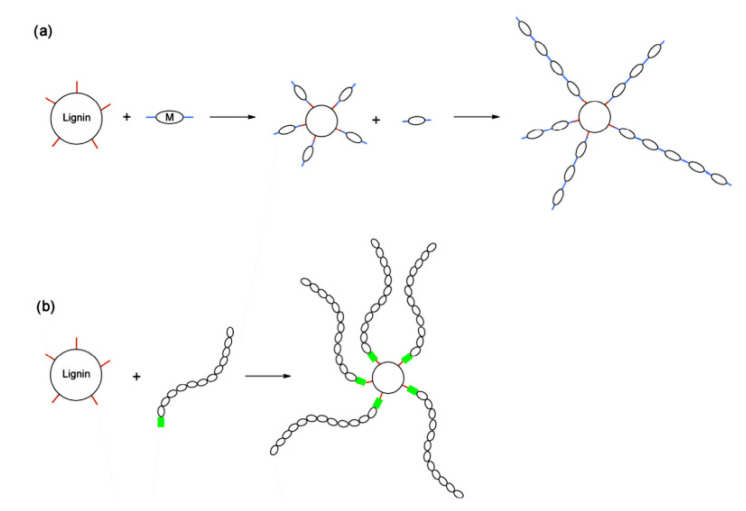
Schematic synthesis of lignin graft copolymers by “grafting from” (**a**) and “grafting onto” (**b**) techniques [1].

**Figure 12 polymers-12-01795-f012:**
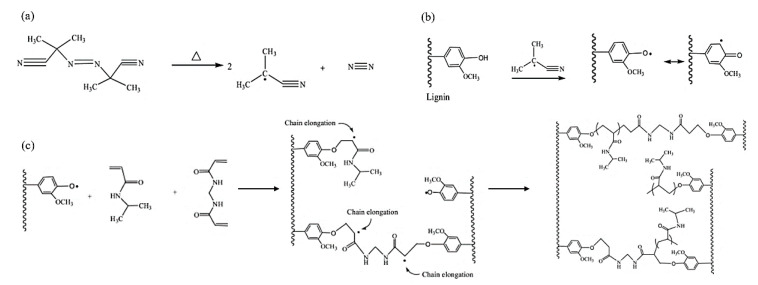
Radical polymerization reaction for lignin-based hydrogel production: (**a**) decomposition of Azobisisobutyronitrile (AIBN) initiator, (**b**) formation of phenoxy radicals, and (**c**) cross-linking reaction [99].

**Figure 13 polymers-12-01795-f013:**
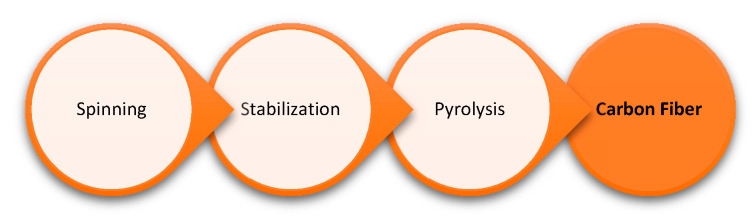
Workflow for the use of lignin as a precursor for carbon fiber.

**Figure 14 polymers-12-01795-f014:**
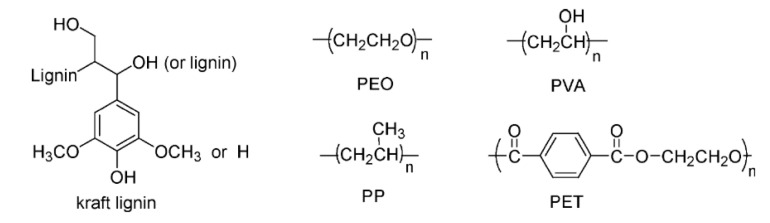
Structural representations of the various polymeric materials that can be blended with lignin [45].

**Figure 15 polymers-12-01795-f015:**
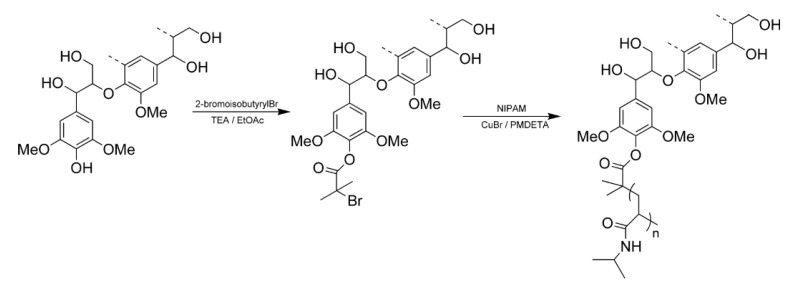
Scheme for the Preparation of Lignin-*g*-polyN-isopropylacrylamide (NIPAM) Copolymers [124].

**Table 1 polymers-12-01795-t001:** Frequency of lignin linkages in native hardwood species.

Biomass	Linkages (%)						References
	β-O-4	β-5	α-O-4	β-β	5-5	4-O-5	
Hardwood	60–62%	3–11%		3–12%	<1%	2%	[38]

**Table 2 polymers-12-01795-t002:** Frequency of lignin linkages in hardwood kraft lignin.

Lignin	Linkage (per 100 Ar)			
	β-O-4	β-5	β-β	References
Birch	2	2	3	[39]
*E. globulus*	12	1	2	[39]
*E. grandis*	5	2	3	[39]

**Table 3 polymers-12-01795-t003:** Molecular weight distribution of hardwood kraft lignins.

Sample	M_w_ (g/mol)	M_n_ (g/mol)	PDI	Method	Ref.
HWKL	3900	1000	3.9	HPLC	[42]
HWKL	3300	1000	3.3	HPLC	[42]
*Eucalyptus exserta*	55,500	31,700	1.8	MALLS	[40]
Indulin	3700	1300	2.3	GPC	[43]
Eucalin	3900	1700	2.3	GPC	[43]
Ligflow 401	2042	719	2.8	GPC	[44]
HWKL	2400	1263	1.9	GPC	[45]
HWKL	3290	1793	1.8	GPC	[46]
European beech	1711	1044	1.6	GPC	[47]
Eucalyptus	4200	1273	3.3	HPLC	[48]
*Eucalyptus grandis*	1740	910	1.9	GPC	[49]

**Table 4 polymers-12-01795-t004:** Functional groups in hardwood kraft lignins.

Sample	Functional Groups					Ref.
	Methoxyl	Phenolic-OH	Aliphatic-OH	Carboxyl	Carbonyl	
*Eucalyptus* spp.	18.8%	3.16 mmol/g	1.54 mmol/g	-	-	[48]
HWKL	-	1.93 mmol/g	1.56 mmol/g	0.39 mmol/g	-	[65]
HWKL	5.9 mmol/g	4.3 mmol/g	4.1 mmol/g	-	-	[66]
HWKL	-	1.45 meq/g		2.23 meq/g	0.13 meq/g	[67]

**Table 5 polymers-12-01795-t005:** Elemental composition of hardwood kraft lignins (wt%).

Sample	C	H	O	N	S	Na	Ash	Ref.
Eucalyptus	63.2	5.1	-	1.7	1.4	1.2	3.2	[74]
European beech	55.7	4.6	31.7	0.3	3.9	-	3.8	[47]
HWKL	62.5	5.7	29.0	0.3	2.9	-	0.5	[73]

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
