# Peer review of "Insights into the Potential of Hardwood Kraft Lignin to Be a Green Platform Material for Emergence of the Biorefinery"

_polymers, 2020, doi:10.3390/polym12081795_

Round 1

Reviewer 1 Report

The present manuscript is interesting and explores the fractionation of the lignin until become a raw material for a particular application. However, I would like to suggest some changes to complete/improve the manuscript:

  • Is important clarify the difference between hardwood and softwood, with examples.
  • Is necessary more uniformity along the text, because sometimes authors write in full words, others put it in abbreviation.
  • It would be important to discuss ionic liquids and eutectic solvents.

In addition there are articles that would be pertinent to be added to the present manuscript. Examples are:

  • Brief Overview on Bio-Based Adhesives and Sealants.
  • Lignin Modification Supported by DFT-Based Theoretical Study as a Way to Produce Competitive Natural Antioxidants.
  • Study on the residual lignin in Eucalyptus globulus sulphite pulp.

Author Response

Please see soft copy.

Reviewer 2 Report

This manuscript reviews the potential use of hardwood kraft lignin, including extraction processing procedure, structure and properties, fractionation, modification and applications. Overall it is well-written and I find only some small problems:

  1. Line 156: "The magnitude of Mw and Mn results depends on the technique used for their determination;overall, values from light scattering are much larger than those obtained from conventional protocols (HPLC and GPC) [33]." This statement is not correct. The value of molecular weight does not depend on the techniques, on the contrary, the techniques used to characterize materials depend on the molecular weight and size. For example, light scattering can only be used for big samples, otherwise, the samples can't be detected. 
  2. Line 435: "as it possessed smaller surface area and more porous structure than synthetic hydrogels [87]" should be less porous structure.
  3. Line 255 "Although lignin presents potential direct applications in the polymer industry, it can only be incorporated in small amounts, considering its thermal degradation and mechanical properties." In the previous section 3, thermal degradation and mechanical properties were not included, probably these two properties could be added. 
  4. Grammar error or typos:   Line 35, 47, 130

Author Response

Please see soft copy.
